# Impact of financial literacy, mental budgeting and self control on financial wellbeing: Mediating impact of investment decision making

**Ruofan Bai** *

School of Accountancy, Henan Institute of Economics and Trade, Zhengzhou, Henan, China

* brfhn@126.com

**Data Availability Statement:** All relevant data are within the paper and its Supporting Information files.

**Funding:** The author(s) received no specific funding for this work.

## Abstract

The topic of financial wellbeing is a current concern within the realm of personal and household finance. This study aims to examine the influence of cognitive factors, specifically financial literacy, mental budgeting, and self-control, on subjective financial wellbeing. While there exist multiple determinants of financial wellbeing, this research focuses on these particular cognitive factors. The present study aims to examine the mediating role of investment decision-making behavior in the association between cognitive factors and financial wellbeing. The study employed Partial Least Squares Structural Equation Modeling (PLS-SEM) to analyze the data collected from a sample of 449 Chinese university students, with the aim of assessing the empirical associations. The results indicate that financial literacy, mental budgeting, and self-control exert a favorable and noteworthy influence on an individual's financial well-being. The results indicate that individuals with a greater degree of financial literacy are more prone to achieving superior financial well-being. Moreover, individuals who practice mental budgeting, a technique that entails mentally classifying and monitoring their expenditures, demonstrate elevated levels of financial well-being. Likewise, the exercise of self-regulation is identified as a pivotal element that impacts an individual's financial wellbeing. The findings indicate that there is evidence to support the mediator, investment decision-making behavior. This mediator partially mediates the association between the independent variables, namely financial literacy, mental budgeting, and self-control, and financial well-being. The results suggest that individuals with elevated levels of financial literacy, proficient mental budgeting skills, and self-regulatory abilities are inclined towards demonstrating favorable investment decision-making conduct. Consequently, this contributes to their general financial welfare. In general, the study's theoretical implications augment the current knowledge repository, while its practical implications provide feasible perspectives for policymakers, financial institutions, and individuals to foster financial wellness and enhance financial results.

**Competing interests:** The authors have declared that no competing interests exist.

## 1. Introduction

The world's biggest public health issues are depression, anxiety, and stress. Mental health issues affect 84 million Europeans, or 17.3% of the population, according to the Global Health Data Exchange [1]. Depression and anxiety are the most common mental health problems in society. Public health concerns include anxiety and mood disorder increase [2]. In 2017, Fiksenbaum et al. [3] discovered that 4.4 percent of people worldwide have anxiety disorders, with 3.6 percent exhibiting symptoms. Poor mental health is caused by psychological stress, employment issues [4], and socioeconomic causes [5]. Financial hardship is rising, according to WHO data [6]. The UK survey [7] and PwC's Employee Financial Wellness Survey [8] in the US reached similar conclusions. Employees reported financial stress higher than any other type of stress in the poll. The Chinese social structure has similar issues. HSBC found that 64% of Chinese are satisfied with their finances in 2019, up from 57% in 2016. The poll found that 36% of Chinese people worry about saving for retirement and 29% about paying unexpected costs [9]. Researching a person's money management, spending, saving, and investing habits is called "financial well-being" [10]. Lack of financial wellbeing causes financial distress, which lowers physical and mental health and workplace productivity [11–14]. If they think their salary isn't enough to meet their basic needs, people feel disadvantaged financially [15].

Several things affect and improve financial well-being. Financial literacy helps people make informed and responsible financial decisions, boosting financial stability and reducing financial worries [16]. Financial literacy gives people the information, skills, attitudes, and behaviors to manage their money and reach their goals [17]. It educates people about financial ideas, risks, and decisions [18]. Financial literacy encourages budgeting, saving, and investing, which improves financial health. They can handle complex financial decisions and avoid financial hazards better. Financial literacy helps people manage their income at difficult times like the COVID-19 epidemic [19]. Mental budgeting is also crucial to financial health. Individuals and households utilize cognitive operations to arrange and control their finances [20]. It entails budgeting for several expenditure categories and mentally dividing the monies [21]. Mental budgeting aids in spending tracking, goal setting, and financial decision-making [20]. Many research have shown the benefits of mental budgeting for financial health. According to Chun & Johnson [22], consumers with superior mental budgeting skills are more resistant to store promotions and price fluctuations. This implies that mental budgeting can help people avoid impulse purchases and stay to their budgets. Financial well-being also depends on self-control. Many research have examined the relationship between self-control and financial outcomes like financial assets and financial management behavior. Self-control is the ability to manage ideas, emotions, and actions to attain long-term goals and avoid temptation [23]. It requires the ability to think rationally, control impulses, and manage money [23, 24]. Higher self-control has been linked to improved financial outcomes. Better self-control leads to increased financial assets [24]. They are also more likely to budget, save, and regulate spending [25]. Self-control improves financial planning and saving [26]. Greater self-control is a key predictor of financial security, as persons with it tend to save and avoid debt [27].

Investment decisions are vital to financial well-being. Several elements increase financial well-being through investing decision-making. Financial literacy matters. According to Kamakia et al. [28], financially literate people make better investment decisions and have higher financial stability and well-being. Financial literacy improves investment decisions by helping people understand and analyze information [29]. Investment decision-making mediates mental budgeting and financial well-being [22]. Financial decision-making is influenced by mental budgeting. Credit cards can mix expenditures across budgeted categories and increase temporal distance between purchases and payments, making it harder to remember how much was

spent on each [22]. This may lead to overspending or bad budget management, affecting finances. Self-control affects financial well-being through investment decisions. Self-control is the ability to manage behavior and make decisions that support long-term goals [30]. Research shows that self-control improves financial decisions. Higher self-control leads to more wise investment decisions and improved financial wellbeing [31].

Amid the Great Recession and COVID-19 pandemic, individuals, families, legislators, financial service providers, and financial educators need more understanding about financial wellbeing and how to improve it. This study makes significant literary contributions. First, this study examines the complex interaction between financial literacy, mental budgeting, self-control, and investment decision making, adding to financial wellbeing research. Examination of these structures together provides a more complete knowledge of financial wellbeing variables. Second, study examines mental budgeting and self-control in financial decision making to connect psychological and economic aspects. This integrated approach adds depth to the research by acknowledging that cognitive and behavioral processes affect financial wellbeing as well as economic considerations. Third, investment decision making as a mediator between financial literacy, mental budgeting, self-control, and financial health is another theoretical contribution. This mediation model shows how various factors affect financial well-being. It explains how financial knowledge, mental budgeting, and self-control affect financial outcomes. Financial educators, counselors, and policymakers can apply the study's conclusions. By recognizing investment decision making as a mediator, it suggests interventions and education to improve financial literacy, mental budgeting, self-control, and financial health. The study emphasizes the need to evaluate many financial wellbeing elements at once. It promotes a holistic approach that emphasizes the interdependence of financial literacy, cognitive processes (mental budgeting), behavioral attributes (self-control), and investment decisions in financial well-being.

Rest of the paper is distributed among four sections: literature and hypotheses, methodology, results and conclusions.

## 2. Literature and hypotheses

The body of research that is now available on the topic of financial well-being hints that the concept of financial well-being is a subjective evaluation of one's present and future financial situation [32–35]. The relevance of objective economic measurements, such as a consumer's income, savings, and investments, credit score, credit card debt, regular mortgage payment, and tax payments, was stressed in much early academic research in the financial wellbeing field [36–38]. The subjective evaluation of financial wellbeing, on the other hand, focuses on the consumer's self-assessment of his or her disposition, attitude, belief, and behaviors linked to money management [32, 35]. According to this subjective interpretation of financial wellbeing, two people with comparable salaries or debt loads may regard their own financial wellbeing very differently. Due to importance of subjective financial wellbeing for researchers studying consumer behavior, financial institutions (FIs), non-profit organizations, businesses, and decision-makers in the government, study choose to investigate subjective side of this contrast. The relationships described in the study, which examines the impact of financial literacy, mental budgeting, and self-control on financial wellbeing with the mediating role of investment decision making, can be supported by several theories from the fields of economics, psychology, and behavioral economics, this study uses cognitive dissonance theory. Cognitive Dissonance Theory suggests that individuals strive for consistency in their beliefs and behaviors [39]. Financial literacy, mental budgeting, and self-control can influence the alignment of an individual's financial decisions with their overall financial goals and values, reducing cognitive dissonance and enhancing financial wellbeing.

## 2.1. Financial literacy

Financial literacy refers to the knowledge of basic financial concepts, the ability to apply financial knowledge and skills in managing financial resources effectively, and the ability to make informed financial decisions to achieve financial welfare over a lifetime [40–44]. It involves understanding of financial matters, the ability to make conscious choice of financial products and services, and techniques for making appropriate financial decisions. Financial literacy translates into prosperity and sustainable development and helps in ensuring the financial sustainability of individuals, families, enterprises, and national economies [45]. It also includes a capacity and confidence to handle personal funds appropriately, short-term decision making and solid long-term financial thinking [46]. Moreover, being familiar with finance-related issues and making rational financial decisions based on basic financial knowledge are also crucial components of financial literacy [45, 47]. Subjective financial knowledge, which refers to individuals' self-evaluation of their financial knowledge, has been found to be a stronger predictor of financial behavior and subjective financial wellbeing than objective financial knowledge [48]. This indicates that individuals who perceive themselves to have higher financial knowledge tend to have higher levels of financial satisfaction and overall financial wellbeing. However, Balasubramnian and Sargent [49] investigate gaps between objective financial literacy and self-reported (perceived) financial literacy and report that individuals with high objective financial literacy make better financial decisions. A study by Joo and Grable [50] sought to identify the variables that affect financial contentment. According to the survey's findings, financial contentment is directly influenced by factors including education level, financial literacy, risk, financial capability, financial activity, and financial demands. The findings demonstrated that improving financial behaviors increases levels of financial happiness at high knowledge and skill levels. As a result, their research suggested that financial literacy affected financial well-being directly. Another study [51] looked at the connections between 3,121 clients of a financial consulting firm's financial activity, financial well-being, and health. According to their findings, those who have a greater level of financial well-being are less stressed, more motivated to manage their money, have better family relationships, and are physically and mentally healthier. Due to their advanced age and high level of vulnerability, retirees place a high priority on their financial well-being. They might experience physical or mental health effects from certain financial stress. In a research measuring financial literacy [52], authors found that even those with the information and skills to use that knowledge may not always behave as expected or experience advances in financial well-being due to a variety of factors. Such effects might be caused by cognitive biases, issues with self-control, family, economic, and institutional factors. However, another research [53] discovered that students' perceptions of their financial well-being were significantly influenced by their financial literacy. Higher financial literacy correlates with greater financial well-being, according to a study [16] on financial literacy, financial well-being, and financial concerns. As a result, financial literacy is required to achieve financial well-being.

## 2.2. Mental budgeting

Mental budgeting is the cognitive process that people use to organize, evaluate, and keep track of financial activities [54]. It is a financial management technique that involves categorizing and monitoring expenses and income on a mental level [20]. Mental budgeting has an essential role to play in improving financial well-being because it can positively influence personal financial management [21] and consumer budgeting behavior [22]. Studies have shown that mental accounting can aid in monitoring personal spending, consumption, and investments and improve financial self-efficacy and control [55]. It can help socially excluded individuals

make better financial decisions [56]. Mental accounting also affects budgeting, investing, and spending decisions [57]. Thus, it plays a central role in improving financial health and helps individuals, communities, and governments in managing their finances [58]. Mental budgeting helps individuals manage their finances better, make informed financial decisions, reducing financial stress, and improving financial self-efficacy. According to [59], financial literacy empowers individuals with knowledge and skills to manage their money effectively. Studies have shown that mental budgeting motivates and positively affects personal financial management [20] and reduces unduly risky personal investment behavior by triggering mental budgeting thoughts [21]. The impact of financial wellbeing and mental health are interlinked, and financial stress is a significant source of stress for many individuals, leading to mental health challenges [22]. Mental budgeting has been identified as a key factor in promoting financial wellbeing and reducing the risk of financial stress impacting an individual's mental health. Multiple studies have been conducted to explore the relationship between mental budgeting and financial wellbeing in recent years. A systematic review by [60] identified the importance of proactive prevention, such as financial education and literacy, in reducing the burden of mental depression caused by financial stress. Similarly, mental budgeting was highlighted as a significant factor in promoting positive financial management behaviors, reducing financial stress, and improving financial wellbeing [61].

## 2.3. Self-control

Self-control, often called self-regulation, is the ability to control one's conduct and reduce impulsivity [62]. Research shows that self-control and financial knowledge improve financial well-being [63]. Self-control, financial understanding, and financial literacy affect financial behavior and decision-making [64]. Self-control is needed to manage finances and prioritize goals in personal financial planning programs. A study on self-control, money attitude, and personal financial planning indicated that self-control affects financial planning [65]. Other research have linked self-control to occupational stress [66], self-directed learning readiness [67], and self-disgust [68].

The financial conduct of all different kinds of economic actors can be influenced by one's level of self-control. According to Thaler and Shefrin [69], the concept of self-control may be applied to the individual as if they were an organization. According to Baumeister [70], people have a tendency to get confused as a result of conflicts between their behaviors and feelings; yet, inner strength creates self-control. The research conducted by [71] utilized three aspects of self-control: planning, monitoring, and commitment. The researchers came to the conclusion that self-control has a significant correlation with household net wealth and financial hardship. Self-control is beneficial for making decisions, having a strong will, and achieving success in the future, whether that achievement be being wealthy or prominent. The inability to exercise self-control can result in illogical decision making, a lack of confidence, and disastrous behavioral outcomes. The ability of a person to exercise self-control in the present and make sound choices will determine their potential financial well-being in the future. People tend to put their goals off till later, and when they want to improve their performance, they will sometimes try to restrict their behavior by placing stringent restrictions and deadlines on themselves.

According to [72], people who have deadlines that are too stringent tend to have less self-control than those who have deadlines that are not stringent enough. The difficulty of exercising self-control may also be understood through Shefrin and Thaler's [73] Behavioral Life-Cycle (BLC). According to the BLC theory, the majority of individuals are preoccupied with the challenges and rewards of the now rather than the advantages of the long term. People create mental accounts in order to employ the resources that are accessible to them by

categorizing their wealth into three categories, such as their present income, their current assets, and their future income [74]. According to Moffitt et al. [75], individuals lack control over their income and as a result spend more money on their immediate need rather than putting away more money for retirement and other future needs. People who have higher self-control also have better financial conduct and are able to take excellent care of their financial resources. Self-control is a key factor in both of these areas. They invest their resources in the most effective way possible [76]. They do not waste money on activities or products that are not important to their lives. People who have mastered the art of self-control have been at the forefront of society for eons, and they continue to do so now. This is due to the fact that self-control is a prerequisite for making sage choices and enjoying improved material circumstances. Households who have established saving guidelines save significantly more money than households that lack self-control. According to Kahneman [77], people who have cognitive capacities always manage their money in a way that allows them to attain their objectives and pay for their predictable costs. Planners and doers are the two types of people that Thaler and Shefrin [69] have determined people to be based on how well they exercise self-control over their finances. To them, planners are concerned with the utility across a lifetime, whereas doers are self-centered, shortsighted, and only exist for a short period of time. To live a prosperous and healthy life, both financially and emotionally, one of the goals that one must achieve is to have good financial well-being. This can only be accomplished via exercising self-control. Most of the studies measured self-control using Brief Self-Control Scale [78] and Short-Term Future Orientation Scale [59].

## 2.4. Investment decision making

Investment decision-making behavior refers to the process of making decisions related to finances and investments by individuals, which are influenced by various factors such as financial knowledge, financial attitude, financial behavior, self-control, psychological biases, and external environment. Financial knowledge plays an important role in making informed financial decisions, while financial behavior refers to the habits and behavior of individuals when managing finances. Self-control enables individuals to make rational and informed decisions while managing their finances. Psychological biases such as herding, heuristics, and prospect also affect the financial decision-making behavior of individuals. In summary, financial decision-making behavior is a complex process influenced by various rational and psychological factors that impact an individual's financial wellbeing [79–82].

Financial or investment decision making behavior is a crucial determinant of financial well-being. It has been established that this behavior has a positive influence on financial well-being [83]. Moreover, financial well-being is directly and indirectly related to financial behavior [83]. Financial behavior is the result of putting expectations and values into action, and it is the link between expectations and financial well-being. Hence, better financial behavior translates to better financial well-being.

Several studies have shown that financial literacy and self-control are significant determinants of financial behavior and financial well-being. Research has found that financial literacy has a significant direct impact on financial well-being, and it affects financial well-being through financial behavior [84]. Similarly, financial self-efficacy and financial literacy positively influence financial well-being through financial behavior mediation [63].

Furthermore, the research has shown a positive relationship between parental financial socialization and financial literacy, financial behavior, and financial well-being. Delafrooz and Paim [85] found that higher levels of financial literacy led to better financial behavior, which in turn resulted in higher financial well-being. Studies have also explored the relationships

between financial behavior, financial knowledge, and financial well-being. For instance, research [86, 87] showed that subjective knowledge had stronger relationships with both financial behavior and financial well-being than objective knowledge. Further, it was established that money attitudes and financial knowledge significantly influenced financial behavior. Money attitudes have also been found to have a positive influence on financial management behavior, which in turn impacts financial well-being.

In conclusion, financial decision making behavior has a significant impact on financial well-being. Financial literacy, self-control, parental financial socialization, financial knowledge, and money attitude have been shown to influence financial behavior and thus impact financial well-being. It is crucial, therefore, to educate individuals on the importance of financial behavior and its role in achieving financial security.

Based on previous discussion, following hypotheses are developed:

H1: financial literacy has a significant direct impact on financial wellbeing.

H2: Mental budgeting has a significant direct impact on financial wellbeing.

H3: Self-control has a significant direct impact on financial wellbeing.

H4: Investment decision making has a significant direct impact on financial well-being.

H5: Investment decision making has a significant mediating effect between mental budgeting and financial well-being.

H6: Investment decision making has a significant mediating effect between financial literacy and financial well-being.

H7: Investment decision making has a significant mediating effect between self-control and financial well-being.

The conceptual model represents the selections of the variables from the critical review of the literature, and we expect their relationship in shape of figure. Moreover, our conceptual model of the study is given in Fig 1.

## 3. Methodology

### 3.1. Data

Sample was chosen using the criteria based on number of items. Convenience sampling was used to collect the data. Data was collected from Chinese university students using both physical and electronic channels which resulted in a set of 449 useable observations. Respondents included 245 male (55%) and 204 female (45%) students. 270 (60%) of these respondents belonged to business major. Table 1 shows the distribution of collected data.

### 3.2. Measures

We employed two separate measures to capture the diverse aspects of one's financial well-being: one, the extent to which one suffered from financial anxiety, and the other is the degree to which one felt financially secure. For the purpose of quantifying anxiety caused by financial concerns, four items from [88] were selected. When calculating the level of financial security, [14] takes into account three different factors. The respondent was asked to rate how strongly they agreed or disagreed on a scale from 1 (strongly disagree) to 5 (strongly agree).

Seven items for measuring financial literacy have been adopted from a study [89].

Four items were adopted from a past study of mental budgeting andmanagement of household finance [59].

Self-control is quantified through a general measure which is a smaller version of the Brief Self-Control Scale [78]. It consists of five items, and the four items from the Short-Term Future Orientation Scale [59].

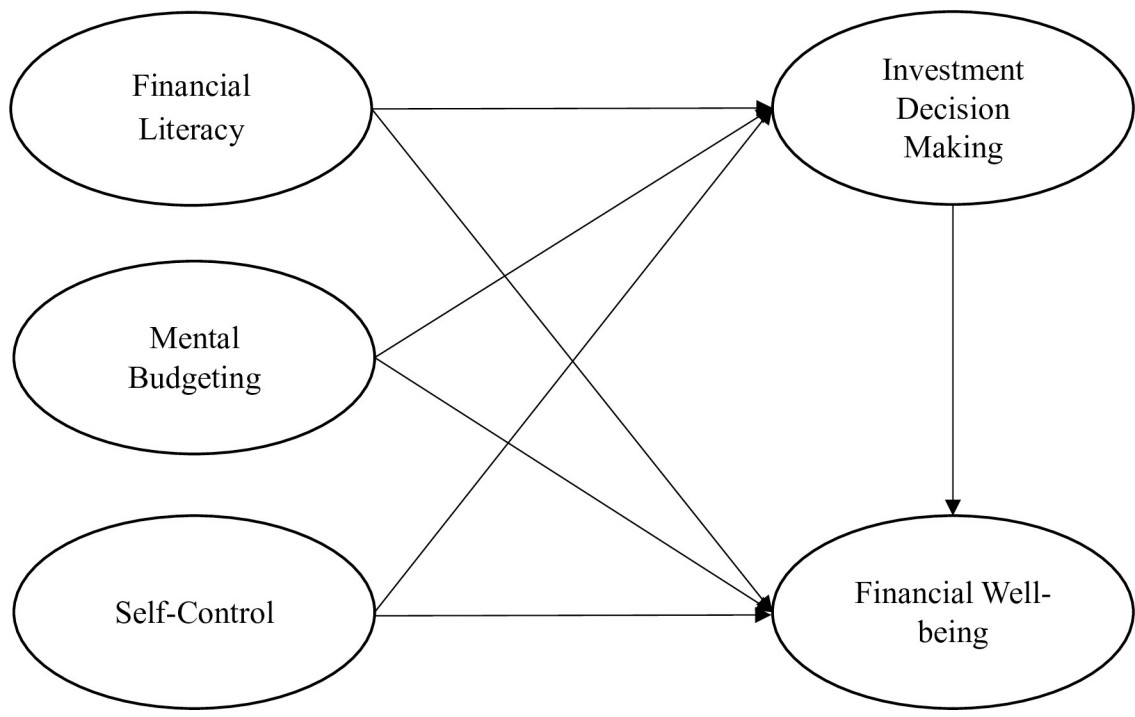

**Fig 1. Research conceptual model.**

Scale for financial management or investment decision making behavior is adopted [90] and contains four components: overall financial management or decision making behavior, savings and investment, cash management and credit management.

List of the items used for measurement is given in S1 File.

### 3.3. Ethical statement

The present investigation pertains to the participation of human subjects, and therefore, ethical clearance was obtained subsequent to its evaluation by the research council of Henan Institute of Economics and Trade, located in Zhengzhou, Henan, China. The study was conducted in accordance with the research ethics guidelines of Henan Institute of Economics and Trade. Participants were accurately informed what is being studied, the benefits and risks of the study. Participants were also aware of their right to withdraw from the study at any point, all respondent gave their verbal informed consent for inclusion before they participated in the study.

**Table 1. Data.**

| Variable | Indicator | Number | Percentage |
| --- | --- | --- | --- |
| Gender | Male | 245 | 54.6 |
| | Female | 204 | 45.4 |
| Ongoing Qualification Level | Undergraduate | 239 | 53.2 |
| | Masters | 112 | 24.9 |
| | Doctor | 98 | 21.8 |
| Ongoing Qualification Major | Business | 270 | 60.1 |
| | Non-Business | 179 | 39.9 |

### 3.4. Analysis

Path analysis and regression are two areas in which Structural Equation Modeling (SEM) excels. SEM is particularly useful when dealing with several variables. The PLS-SEM approach is being utilized throughout this investigation so that Path analysis may be performed. The benefits of utilizing PLS-SEM include the fact that it is more flexible with the sample sizes and is also less vulnerable to the violations of the multivariate data assumptions, such as normality of data. These are only few of the advantages of utilizing PLS-SEM. [91].

## 4. Results

### 4.1. Measurement model

Stage one in the estimation of the measurement model included indicator reliability measurement through factor (outer) loadings, internal reliability measurement through composite reliability, convergent validity measurement through average variance extracted. The table shows that all of the indicators have loadings of more than 0.50 (range: 0.637–1.000), which is the value recommended by Nunnally [92] and Hair et al. [93]. All of the constructs obtained composite reliability values (range: 0.714 to 0.845) greater than 0.70, which is the value recommended by [94–96]. Table 2 shows that the Cronbach's alpha ranges from 0.743 to 0.821; the statistically acceptable minimum value is 0.70 [94–96]. AVE values should be greater than 0.50 [93, 97]. Our results meet these criteria.

### 4.2. Structural model

The path coefficients were assessed in order to test the hypotheses and determine the association between the psychological characteristics of the young people and their financial conduct as well as their overall financial well-being. The value of a variable's path coefficient indicates the extent to which that variable was directly influenced by another variable. A value that is closer to 1 indicates that there is a stronger correlation, while a value that is closer to 0 indicates that there is a weaker relationship. Values close to zero are not statistically significant. Path coefficients are listed in the Table 3. Results indicate that all three independent variable (financial literacy, mental budgeting and self-control) are positively affect the dependent variable (financial-wellbeing).

### 4.3. Mediation

Baron and Kenny [98] argued for simultaneously considering direct and indirect effects to conclude mediation tests. We found that the direct effects of the independent variables (financial literacy and investment decision making behavior) on the dependent variable (financial well-being) were positive and statistically significant. Similar is the case for other two paths i.e. mental budgeting and investment decision making behavior and self-control and investment decision making behavior. The indirect effects in the presence of the mediator (investment decision making behavior) is also statistically significant. This concludes into partial mediation. The mediation results are summarized in Table 4.

## 5. Conclusion

Based on this investigation, it is evident that financial literacy, mental budgeting, and self-control have a positive and significant impact on subjective financial well-being. The findings suggest that individuals who possess a higher level of financial literacy are more likely to experience better financial well-being. This implies that having a strong understanding of financial concepts, such as budgeting, saving, and investing, can contribute to improved

**Table 2. Factor loadings, composite reliability, Cronbach alpha, and average variance explained.**

| Variable | Items | Loadings | CR | Cronbach Alpha | AVE |
|---|---|---|---|---|---|
| Financial Wellbeing | FWB1 | 0.823 | 0.845 | 0.821 | 0.773 |
| | FWB2 | 0.785 | | | |
| | FWB3 | 0.923 | | | |
| | FWB4 | 0.812 | | | |
| | FWB5 | 0.801 | | | |
| | FWB6 | 0.765 | | | |
| | FWB7 | 0.873 | | | |
| Financial Literacy | FL1 | 0.772 | 0.791 | 0.743 | 0.764 |
| | FL2 | 0.704 | | | |
| | FL3 | 0.770 | | | |
| | FL4 | 0.818 | | | |
| | FL5 | 0.880 | | | |
| | FL6 | 0.859 | | | |
| | FL7 | 0.737 | | | |
| Mental Budgeting | MB1 | 0.760 | 0.868 | 0.841 | 0.702 |
| | MB2 | 0.822 | | | |
| | MB3 | 0.855 | | | |
| | MB4 | 0.807 | | | |
| Self-Control | SC1 | 0.776 | 0.799 | 0.796 | 0.692 |
| | SC2 | 0.804 | | | |
| | SC3 | 0.858 | | | |
| | SC4 | 0.715 | | | |
| | SC5 | 0.769 | | | |
| | SC6 | 0.838 | | | |
| | SC7 | 0.810 | | | |
| | SC8 | 0.770 | | | |
| | SC9 | 0.871 | | | |
| Investment Decision Making Behavior | DMB1 | 0.842 | 0.714 | 0.710 | 0.811 |
| | DMB2 | 0.855 | | | |
| | DMB3 | 0.866 | | | |
| | DMB4 | 0.848 | | | |
| | DMB5 | 0.857 | | | |
| | DMB6 | 0.874 | | | |
| | DMB7 | 0.856 | | | |
| | DMB8 | 0.809 | | | |
| | DMB9 | 0.760 | | | |

**Table 3. Path analysis.**

| Paths | Beta | SE | t-stat | p-value | Remarks |
|---|---|---|---|---|---|
| FL --> DMB | 0.136 | 0.028 | 4.871 | 0.000 | Supported |
| MB --> DMB | 0.128 | 0.019 | 6.631 | 0.000 | Supported |
| SC --> DMB | 0.152 | 0.024 | 6.399 | 0.000 | Supported |
| DMB --> FWB | 0.126 | 0.067 | 1.892 | 0.059 | Supported |
| FL --> FWB | 0.299 | 0.037 | 8.073 | 0.000 | Supported |
| MB --> FWB | 0.102 | 0.028 | 3.704 | 0.000 | Supported |
| SC --> FWB | 0.182 | 0.033 | 5.494 | 0.000 | Supported |

**Table 4. Mediation analysis.**

| Path | Direct Effect | p-value | Indirect Effect | p-value | Remarks |
|---|---|---|---|---|---|
| FL --> FWB | 0.136 | 0.000 | 0.017 | 0.034 | Partial Mediation |
| MB --> FWB | 0.128 | 0.000 | 0.016 | 0.035 | Partial Mediation |
| SC --> FWB | 0.152 | 0.000 | 0.019 | 0.036 | Partial Mediation |

financial outcomes. These findings are consistent to previous literature [45–49]. Furthermore, individuals who engage in mental budgeting, which involves mentally categorizing and tracking their expenses, exhibit higher levels of financial well-being. This practice enables them to have a better grasp of their financial situation and make informed decisions regarding their spending habits and financial goals. Previous literature support this finding [22, 56, 60, 61]. Similarly, self-control emerges as a crucial factor influencing financial well-being. Individuals who exercise self-control, such as resisting impulsive purchases and sticking to their financial plans, are more likely to achieve better financial outcomes. This finding suggests that maintaining discipline and self-restraint in financial matters can significantly contribute to one's financial well-being. This outcome is also in line with the existing empirical evidence [63, 64, 69]. Results reveal support for the mediator, investment decision-making behavior, which partially mediates the relationship between the independent variables (financial literacy, mental budgeting, and self-control) and financial well-being. The findings indicate that individuals who possess higher levels of financial literacy, engage in effective mental budgeting, and exercise self-control are more likely to exhibit positive investment decision-making behavior. This, in turn, contributes to their overall financial well-being. The partial mediation suggests that investment decision-making behavior accounts for a portion of the relationship between the independent variables and financial well-being, while other factors may also be involved. These results have important implications for understanding the pathways through which financial literacy, mental budgeting, and self-control influence financial well-being. The presence of mediation indicates that investment decision-making behavior plays a role in translating the effects of these independent variables into improved financial outcomes. It highlights the significance of making informed investment decisions and aligning them with one's financial goals [79–82]. Overall, the results of this investigation underscore the importance of financial literacy, mental budgeting, and self-control in shaping an individual's financial well-being. To enhance financial well-being, individuals should strive to improve their financial knowledge, develop effective mental budgeting strategies, and cultivate self-control in their financial decision-making processes.

This study has several theoretical and practical implications.

## 5.1. Theoretical implications

Enriching the understanding of financial well-being: This study contributes to the existing body of knowledge by providing empirical evidence on the impact of financial literacy, mental budgeting, and self-control on financial well-being. It enhances our theoretical understanding of the factors that influence individuals' financial well-being and highlights the importance of these variables in achieving positive financial outcomes.

Supporting the importance of financial education: The findings underscore the significance of financial literacy in promoting financial well-being. This emphasizes the need for educational institutions, policymakers, and financial institutions to prioritize and promote financial education programs. It highlights the potential benefits of equipping individuals with the necessary knowledge and skills to make informed financial decisions and improve their financial well-being.

Emphasizing the role of behavioral factors: This study highlights the role of behavioral factors, such as mental budgeting and self-control, in shaping financial well-being. It supports the growing body of research that recognizes the impact of psychological and behavioral aspects on financial outcomes. These findings can contribute to the development of theories and frameworks that integrate behavioral economics and finance, providing a more comprehensive understanding of individuals' financial well-being.

## 5.2. Practical implications

Policy interventions and financial education programs: Policymakers can utilize these findings to design and implement effective financial education initiatives that focus on improving financial literacy, promoting mental budgeting practices, and enhancing self-control. These programs can be targeted towards various age groups and socio-economic backgrounds to ensure wider accessibility and inclusivity.

Financial counseling and guidance: Financial institutions and professionals can leverage the insights from this study to provide personalized financial counseling and guidance to their clients. By addressing specific areas of financial literacy, mental budgeting, and self-control, individuals can receive tailored support to enhance their financial well-being and achieve their financial goals.

Development of digital tools and resources: Technology can play a crucial role in improving financial well-being. Based on the findings of this study, the development of digital tools, mobile applications, and online platforms can be tailored to provide financial education, facilitate mental budgeting, and encourage self-control. These resources can provide real-time feedback, personalized recommendations, and practical tips to help individuals manage their finances effectively.

Overall, the theoretical implications of this study contribute to the existing knowledge base, while the practical implications offer actionable insights for policymakers, financial institutions, and individuals to promote financial well-being and improve financial outcomes.

## 5.3. Limitations

The study's findings may be limited by the characteristics of the sample used. The investigation has focused on a specific demographic and geographic sample (i.e. students), which could limit the generalizability of the results to a broader population. Future research could consider using larger and more diverse samples to enhance the external validity of the findings. The study employed a cross-sectional design, which captures data at a specific point in time. This design limitation prevents establishing causal relationships between the variables investigated. To address this limitation, future research could employ longitudinal or experimental designs to assess the causal effects of financial literacy, mental budgeting, self-control, and investment decision-making behavior on financial well-being. The study relied on self-reported measures, which may introduce response biases and social desirability effects. Participants might have provided answers that they believed were expected or socially acceptable rather than reflecting their true behaviors or beliefs. Future studies could consider incorporating objective measures or alternative data sources to enhance the validity of the findings.

## 5.4. Further research directions

Investigating additional mediating and moderating variables could provide a more comprehensive understanding of the relationships between financial literacy, mental budgeting, self-control, investment decision-making behavior, and financial well-being. Factors such as risk tolerance, financial attitudes, social influences, and psychological factors could be explored to

uncover their potential impact on the relationships of interest. Future research could explore the long-term effects of financial literacy, mental budgeting, and self-control on financial well-being. Assessing the sustainability and durability of these effects over time could shed light on the long-term benefits of cultivating these skills and behaviors. Investigating the effectiveness of interventions aimed at improving financial literacy, mental budgeting, and self-control could provide valuable insights. Assessing the impact of educational programs, financial counseling, and interventions on individuals' financial well-being and investment decision-making behavior would help identify the most effective strategies for promoting positive financial outcomes. Exploring the role of cultural and contextual factors in the relationships of interest could offer valuable insights. Different cultures and socio-economic contexts may influence the impact of financial literacy, mental budgeting, self-control, and investment decision-making behavior on financial well-being. Examining these factors would allow for a more nuanced understanding of how these relationships manifest across diverse populations. Addressing these limitations and pursuing these research directions can further advance the knowledge and understanding of the impact of financial literacy, mental budgeting, self-control, investment decision-making behavior, and their interrelationships on financial well-being.

## Supporting information

**S1 File. Represents the variables items measurement.**
(DOCX)

## Acknowledgments

This paper is a general project of 2021 Henan Higher Education Teaching Reform Research and Practice Project (Research and Practice of "Integration of Competition and Teaching" in Accounting major under the background of National Vocational College Skills Competition 2021SJGLX826)

## Author Contributions

**Conceptualization:** Ruofan Bai.

**Data curation:** Ruofan Bai.

**Formal analysis:** Ruofan Bai.

**Funding acquisition:** Ruofan Bai.

**Investigation:** Ruofan Bai.

**Methodology:** Ruofan Bai.

**Project administration:** Ruofan Bai.

**Resources:** Ruofan Bai.

**Software:** Ruofan Bai.

**Supervision:** Ruofan Bai.

**Validation:** Ruofan Bai.

**Visualization:** Ruofan Bai.

**Writing – original draft:** Ruofan Bai.

**Writing – review & editing:** Ruofan Bai.

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
