## [Decision Letter · Decision Letter 0]

17 Jul 2023

PONE-D-23-18334Impact of Financial Literacy, Mental Budgeting and Self Control on Financial Wellbeing: Mediating Impact of Investment Decision MakingPLOS ONE

Dear Dr. Ruofan Bai,

Thank you for submitting your manuscript to PLOS ONE. After careful consideration, we feel that it has merit but does not fully meet PLOS ONE’s publication criteria as it currently stands. Therefore, we invite you to submit a revised version of the manuscript that addresses the points raised during the review process.

The paper investigate the Impact of Financial Literacy, Mental Budgeting and Self Control on Financial Wellbeing: Mediating Impact of Investment Decision Making". Its findings are interesting but requires major revisions before it can be considered. My comments are as follows:

You need state clearly the contributions of the paper. For example, "Consequently, the current paper seeks to make the following contributions to the existing literature. First,…, Second,…., Third, …, Fourth,… and so on". The description of the contribution needs to be more forensic, needs to be more focussed.The authors should discuss the relevant theories in detail and relate their findings to of financial literacy, mental budgeting, and self-control on Financial Wellbeing.Highlight their economic and research and policy implications. In the discussion of the results please focus on the novel findings and insights vis-à-vis the existing literatureTheoretical framework may increase its implication, Read the below related paper for methodology.

R. M. Ammar Zahid.,Rafique., S. Khurshid., M. Khan., W. Ikram Ullah (2023). Does women’s Financial Literacy accelerate Financial Inclusion? Evidence from Pakistan. *Journal of the Knowledge Economy*, DOI:https://doi.org/10.1007/s13132-023-01272-2

I recommend Major REVISIONS for publication after the author/s addressing the above queries and suggestions.

The paper investigate the Impact of Financial Literacy, Mental Budgeting and Self Control on Financial Wellbeing: Mediating Impact of Investment Decision Making". Its findings are interesting but requires major revisions before it can be considered. My comments are as follows:

You need state clearly the contributions of the paper. For example, "Consequently, the current paper seeks to make the following contributions to the existing literature. First,…, Second,…., Third, …, Fourth,… and so on". The description of the contribution needs to be more forensic, needs to be more focussed.The authors should discuss the relevant theories in detail and relate their findings to of financial literacy, mental budgeting, and self-control on Financial Wellbeing.Highlight their economic and research and policy implications. In the discussion of the results please focus on the novel findings and insights vis-à-vis the existing literatureTheoretical framework may increase its implication, Read the below related paper for methodology.

R. M. Ammar Zahid.,Rafique., S. Khurshid., M. Khan., W. Ikram Ullah (2023). Does women’s Financial Literacy accelerate Financial Inclusion? Evidence from Pakistan. *Journal of the Knowledge Economy*, DOI:https://doi.org/10.1007/s13132-023-01272-2

I recommend Major REVISIONS for publication after the author/s addressing the above queries and suggestions.

We look forward to receiving your revised manuscript.

Kind regards,

Wajid Khan

Academic Editor

PLOS ONE

Journal Requirements:

Reviewers' comments:

Reviewer's Responses to Questions

**Comments to the Author**

1. Is the manuscript technically sound, and do the data support the conclusions?

Reviewer #1: No

Reviewer #2: Yes

Reviewer #3: Partly

2. Has the statistical analysis been performed appropriately and rigorously? 

Reviewer #1: Yes

Reviewer #2: No

Reviewer #3: Yes

3. Have the authors made all data underlying the findings in their manuscript fully available?

Reviewer #1: Yes

Reviewer #2: Yes

Reviewer #3: Yes

4. Is the manuscript presented in an intelligible fashion and written in standard English?

Reviewer #1: Yes

Reviewer #2: Yes

Reviewer #3: Yes

5. Review Comments to the Author

Reviewer #1: Overall: I think this is a major world problem and a significant research topic. I would like to see more than perception self-report measures to make the claims in this paper. Improving actual financial literacy vs. perceived financial literacy (self-report impressions that you have here) may lead to different recommendations. You lack important covariates. You may be able to rework this to be about perceptions influencing other perceptions and that might work. After all, you make the point that perceived FWB is more about the individual’s interpretation, not actual financial status. You have a specific population (students) that may not have had extensive experience with financial decisions. That is a limitation (at least acknowledge it).

Introduction

You may be over playing the contribution to mental health. It is enough to indicate that financial stress is an important stressor. This can be shortened, perhaps substantially. You do not build up to your research questions (such that they are a logical next step in addressing financial well-being). For instance, you never mention mental budgeting or investment decisions prior to line 161 where they appear as a surprise. You have explained the problem but then you need to show how your work will address it. That link needs strengthening. You mention that many studies do not include a full array of variables, which is true, then you drop that aspect and have RQs that are lean (single variable), the very thing you criticize. You then have a paragraph about the importance of increasing FWB with factors not in your RQs (like retirement planning). Create a more concise organization: describe the problem (which is done extensively), how you address the problem (missing, RQs are not really doing the job here), the major findings and the implications given the findings. The specific RQs usually come after hypothesis.

Literature and Hypothesis

Just listing your variables and the literature for that variable is good but needs some introduction prior to jumping into them. Why start with FL? Is this for perceived and actual (measured) FL or just actual FL? In many models they have similar correlations on financial behaviors but mean slightly different things. Not sure what you mean here: “Managing finances for personal needs aligns with

financial literacy from a practical perspective.” (Line 193). Your comment: Financial literacy and financial well-being have been the subjects of separate research up until this point is unclear.” (Line 198) Most of the literature on FL is impact on financial habits or financial outcomes, which you argue leads to FWB. Then you start discussing FWB, leaving FL. This seems like the place to put your RQ about FL.

In the second paragraph about mental budgeting you pivot to FL, (Line 230) and then back to mental budgeting. In line 235 you say mental budgeting has the “most” influence on financial behavior – more than any other variable – with one citation given. I do not agree as I look at Xiao & O’Neill (2018). The beta for that variable is much smaller than several other significant variables in the OLS on financial well-being. Further, other studies have found it to be significant but not with a higher effect size over other variables such as education and income. The discussion on self-control does not mention how it was measured in these studies that found it correlated with positive financial outcomes. Is self-control nudgeable (vs. FL)?

Line 324: FL and self-control are significant in financial behavior and well-being. Seems to be leaving off some major variables (e.g., education, income, demographics). The discussion needs tighter organization. You then pivot to financial education and subjective FL in the mental budgeting section. Then parental education comes in (before individual’s education?) and money attitudes.

I might introduce the full family of variables that link to financial behaviors and FWB and then one at a time discuss them, including how they are measured in the literature, leading to each RQ.

Method and Findings

Please put the results on Fig 1.

Wonder if students are typical of the national studies done in the literature? Most have not even started their careers or thought about emergency savings. I wonder about the ability to generalize from this.

Actual FL has been measured by the same 5 questions for decades and include, for example, the ability to figure compounding interest at 2% for one year. See Lusardi’s line of research. I believe you have measure “perceived financial literacy,” not actual FL. Perceived is the person’s belief about financial competency. This differs from actual competency. For instance, for a discussion of the difference of these two, see:

Balasubramnian, Bhanu, and Carol Springer Sargent. “Impact of Inflated Perceptions of Financial Literacy on Financial Decision Making.” Journal of Economic Psychology, vol. 80, 2020. https://doi.org/10.1016/j.joep.2020.102306.

You have failed to control for very important demographic information, especially education and income. Further, your FL variable, normally measured by an actual “test” of financial comprehension, is a FL perception self-report. Highly educated individuals may have better self-control and so you cannot tell which is giving you the result without controlling for these important covariates. You rightly criticize the literature for not including all the variables and then you do it too.

You may have a finding here but not the one you discuss. You have found that one’s view about financial competence correlates with financial behaviors and financial well-being. And how do we know the direction? Maybe strong financial behaviors (paying things on line for instance) makes the individual feel more able (PFL) and in control (self-control). Could it be the perceptions follow from good habits and perceptions lead to FWB? You mention that two different individuals with the same financial circumstances can have different SWB perceptions. This would be a different paper but may be possible with this dataset.

Going forward, consider a more diverse sample, measuring AFL, and including covariates.

Small items to address:

• I would not characterize the cognitive aspects as a “personality” type. (line 155)

• Line 161: Remove “to”

Reviewer #2: Dear Authors

Why financial literacy and financial knowledge are not operationalized as multi-dimensional constructs.

Financial literacy is measured by both subjective and objective approaches. Why authors adopted subjective approach only.

Why Baron and Kenny's approach of mediation was adopted. Baron and Kenny's approach is outdated an primitive approach of mediation.

Reviewer #3: The paper entitled” Impact of Financial Literacy, Mental Budgeting and Self Control on Financial Wellbeing: Mediating Impact of Investment Decision Making” deals with a very interesting topic and it included interesting ideas. In general, I appreciate the aims of this work; it is quite interesting and informative to most readers of this field.

However, I have the following comments that hopefully help the authors improve their paper:

• The introduction section is too long. The reader is lost with the overwhelming amount of background information that relates to the topic but is not necessarily relevant for your research. It may be wise to remove some paragraphs if they are not strongly related to the main issue.

• The structure (outline) of the paper could be given at the end of the introductory chapter.

• I suggest that the authors add a research method diagram. This will provide a snapshot of the research steps followed and will help the reader in a clearer understanding of the paper.

• In relation to literature review, I would strongly encourage authors to provide a summary table of comprehensive literature review that will not only identify the gaps in the literature but also strengthen the contribution of this work.

• What are the limitations of the study in terms of the proposed method, data used, approaches, and/or analysis?

• How the results of this study can be generalized to other regions?

• The authors should convince the readers, that their contribution is so important. These issues deserve a deeper discussion: What are the managerial implications from this research? How does this understanding help people to make better decisions? How decision or policy makers could benefit from this study.

• As usual a final thorough proof-reading is recommended.

I wish the author(s) all the best for their research and that these comments will be useful to them in improving the paper.

6. PLOS authors have the option to publish the peer review history of their article (what does this mean?). If published, this will include your full peer review and any attached files.

Reviewer #1: No

Reviewer #2: **Yes: **Suhail Ahmad Bhat

Reviewer #3: No

---

## [Author Response · Author response to Decision Letter 0]

18 Sep 2023

RESPONSE TO EDITOR

Contributions are rewritten clearly.

Relevant theory is added.

Implications of the findings are added.

RESPONSE TO REVIEWER 1

Concerns about objectives and subjective financial wellbeing are addressed and it is mentioned that author is interested in the later.

Introduction is shortened.

After discussing the problem, solution (i.e. role of selected factors) is also discussed.

Comments which were not matched as indicated by the reviewer are either removed or adjusted.

Education, income and demographics are ignored because of the sample, as respondents do not have very diverse characteristics.

RESPONSE TO REVIEWER 2

Item 1: Why financial literacy and financial knowledge are not operationalized as multi-dimensional constructs?

Operationalizing "financial literacy" as a uni-dimensional construct can have some advantages. Here are some reasons why author choose to operationalize financial literacy as a uni-dimensional construct:

Simplicity: A uni-dimensional approach simplifies measurement and analysis. It reduces the complexity associated with measuring multiple dimensions of financial literacy, making it easier to administer surveys, collect data, and analyze results.

Ease of Communication: Communicating and interpreting the results of a uni-dimensional financial literacy measure is straightforward. It allows for clearer communication of findings to policymakers, educators, and the general public.

Comparison: Uni-dimensional measures make it easier to compare individuals or groups based on a single scale, facilitating straightforward comparisons between different demographics, regions, or time periods.

Policy Implications: A uni-dimensional measure can be more effective for guiding policy decisions, as it provides a clear overall picture of financial literacy levels within a population. Policymakers may find it easier to target interventions when working with a single metric.

Apart from these reasons, the items chosen to measure financial literacy are broad and fulfill the researcher’s objective.

Item 2: Financial literacy is measured by both subjective and objective approaches. Why authors adopted subjective approach only?

Here's why author choose subjective approach:

Self-assessment and self-awareness: Subjective measures allow individuals to assess their own financial knowledge, skills, and confidence in managing their finances. This self-awareness can be valuable because it helps individuals recognize their own areas of weakness and take steps to improve their financial literacy. It can also serve as a motivation for individuals to seek out financial education and make positive changes in their financial behaviors.

Practicality and cost-effectiveness: Subjective measures are often easier and more cost-effective to implement than objective measures, which may require standardized tests or evaluations by financial experts. Subjective assessments can be administered through surveys or questionnaires, making them accessible to a wider range of people and organizations, including schools, employers, and financial institutions.

Cultural and contextual sensitivity: Financial literacy is not a one-size-fits-all concept. It can vary based on cultural, socioeconomic, and personal factors. Subjective assessments can capture the nuances of an individual's financial knowledge and attitudes, allowing for a more context-specific understanding of their financial literacy.

Focus on behavior and decision-making: Subjective measures often include questions about financial attitudes, behaviors, and decision-making, which are critical components of financial literacy. Understanding how individuals perceive and approach financial choices can provide valuable insights for designing targeted financial education programs and interventions.

Item 3: Why Baron and Kenny's approach of mediation was adopted?

Baron and Kenny's approach to mediation analysis is a widely used and influential method in the field of psychology and social sciences for investigating the mechanisms by which an independent variable affects a dependent variable through an intermediate variable (i.e., the mediator).

Clarity and Transparency: Baron and Kenny's approach provides a clear and step-by-step framework for conducting mediation analysis. It helps researchers systematically test the hypothesized mediation model, making the process more transparent and accessible.

Causal Inference: The approach emphasizes the establishment of causality in mediation relationships. It requires researchers to demonstrate that three conditions are met: (a) the independent variable significantly predicts the mediator, (b) the mediator significantly predicts the dependent variable while controlling for the independent variable, and (c) the direct effect of the independent variable on the dependent variable is reduced or becomes non-significant when the mediator is included in the model. This helps researchers make stronger claims about causality.

Practicality: Baron and Kenny's approach is relatively straightforward to implement. It does not require advanced statistical techniques or specialized software, making it accessible to a wide range of researchers.

Interpretability: The approach provides coefficients that are easily interpretable. Researchers can directly assess the size and significance of the indirect (mediation) effect, which is often of primary interest in mediation analysis.

In summary, Baron and Kenny's approach to mediation analysis is preferred for its simplicity, clarity, and emphasis on causality.

RESPONSE TO REVIEWER 3

Introduction is shortened.

The structure of the paper is given.

Limitations are identified.

Implications are discussed.

---

## [Decision Letter · Decision Letter 1]

2 Nov 2023

Impact of Financial Literacy, Mental Budgeting and Self Control on Financial Wellbeing: Mediating Impact of Investment Decision Making

PONE-D-23-18334R1

Dear Dr. Bai,

We’re pleased to inform you that your manuscript has been judged scientifically suitable for publication and will be formally accepted for publication once it meets all outstanding technical requirements.

Kind regards,

Wajid Khan

Academic Editor

PLOS ONE

Additional Editor Comments (optional):

Reviewers' comments:

Reviewer's Responses to Questions

**Comments to the Author**

1. If the authors have adequately addressed your comments raised in a previous round of review and you feel that this manuscript is now acceptable for publication, you may indicate that here to bypass the “Comments to the Author” section, enter your conflict of interest statement in the “Confidential to Editor” section, and submit your "Accept" recommendation.

Reviewer #2: All comments have been addressed

Reviewer #3: (No Response)

2. Is the manuscript technically sound, and do the data support the conclusions?

Reviewer #2: Yes

Reviewer #3: (No Response)

3. Has the statistical analysis been performed appropriately and rigorously? 

Reviewer #2: Yes

Reviewer #3: (No Response)

4. Have the authors made all data underlying the findings in their manuscript fully available?

Reviewer #2: Yes

Reviewer #3: (No Response)

5. Is the manuscript presented in an intelligible fashion and written in standard English?

Reviewer #2: Yes

Reviewer #3: (No Response)

6. Review Comments to the Author

Reviewer #2: All the issues raised have been addressed by the author/s. the paper is now in a position to be published in the journal

Reviewer #3: The paper has significantly improved as compared to the previous version. Indeed, the authors tried to improve it, and the main the weaknesses are solved.

Thus, in my opinion, the paper is on the borderline recommendable for publication.

7. PLOS authors have the option to publish the peer review history of their article (what does this mean?). If published, this will include your full peer review and any attached files.

Reviewer #2: **Yes: **Suhail Ahmad Bhat

Reviewer #3: No

---

## [Editor Report · Acceptance letter]

6 Nov 2023

PONE-D-23-18334R1 

Impact of Financial Literacy, Mental Budgeting and Self Control on Financial Wellbeing: Mediating Impact of Investment Decision Making 

Dear Dr. Bai:

I'm pleased to inform you that your manuscript has been deemed suitable for publication in PLOS ONE. Congratulations! Your manuscript is now with our production department. 

Kind regards, 

on behalf of

Dr. Wajid Khan 

Academic Editor

PLOS ONE